# Depressive and Anxiety Disorders in Patients with Inflammatory Bowel Diseases: Are There Any Gender Differences?

**DOI:** 10.3390/ijerph20136255

**Published:** 2023-06-29

**Authors:** Elia Fracas, Andrea Costantino, Maurizio Vecchi, Massimiliano Buoli

**Affiliations:** 1Department of Pathophysiology and Transplantation, University of Milan, 20122 Milan, Italy; elia.fracas@unimi.it (E.F.); andrea.costantino@policlinico.mi.it (A.C.); maurizio.vecchi@unimi.it (M.V.); 2Division of Gastroenterology and Endoscopy, Foundation IRCCS Ca’ Granda Ospedale Maggiore Policlinico, 20122 Milan, Italy; 3Department of Neurosciences and Mental Health, Fondazione IRCCS Ca’ Granda Ospedale Maggiore Policlinico, 20122 Milan, Italy

**Keywords:** Crohn’s disease, ulcerative colitis, depressive disorders, anxiety disorders, gender differences

## Abstract

Gender differences were identified in the frequency and clinical presentations of inflammatory bowel disease (IBD) and depressive and anxiety disorders, which are more common in IBD patients than in the general population. The present manuscript provides a critical overview of gender differences in the frequency and clinical course of mood and anxiety disorders in IBD patients, with the aim of helping clinicians provide individualized management for patients. All of the included studies found that IBD patients reported a higher frequency of depressive and anxiety disorders than the general population. These findings should encourage healthcare providers to employ validated tools to monitor the mental health of their IBD patients, such as the Patient Health Questionnaire (PHQ-9). In addition, most studies confirm that women with IBD are more likely than men to develop affective disorders and show that up to 65% of women with IBD have depressive and anxiety disorders. Women with IBD require close mental health monitoring and ultimately a multidisciplinary approach involving mental health professionals. Drug treatment in women should be individualized and medications that may affect mental health (e.g., corticosteroids) should be thoroughly reconsidered. Further data are needed to ensure individualized treatment for IBD patients in a framework of precision medicine.

## 1. Introduction

Inflammatory bowel diseases (IBD), including Crohn’s disease (CD) and ulcerative colitis (UC), are chronic inflammatory conditions characterized by intestinal and extra-intestinal symptoms (e.g., arthritis) associated with poor quality of life, especially during acute exacerbations [1,2]. The course of IBD can be complicated by comorbidities such as immune-mediated illnesses including dermatological, ocular, and articular disease, or conditions characterized by inflammatory dysregulation such as metabolic syndrome or mental disorders [3,4].

Mental conditions, with a predominance of depression and anxiety disorders, are common in patients with IBD. Recent articles have reported a higher risk of IBD in patients with schizophrenia [5] and bipolar disorder [6]. The prevalence of depressive disorders in IBD patients varies from 21% to 25%, while anxiety disorders are seen in 19.1% to 35% of patients with IBD [7]. Compared with the general population, IBD patients are twice as likely to have an affective disorder [8]. In addition, the link between IBD and mental disorders seems to be bidirectional: flares of gastrointestinal disease are associated with an increased risk of anxiety and depression, while conversely, patients with mood disorders seem to be at a higher risk of developing IBD [9,10,11,12]. 

Different mechanisms, including biological and psychosocial factors, have been suggested to explain the vulnerability of IBD patients to affective disorders [13]. IBD and affective disorders share inflammatory dysregulation [14,15]. Depressive disorders, in particular, are exacerbated by cell-mediated immunity alterations similarly to CD [16,17], which would explain the higher prevalence of mood disorders in CD than in UC [18]. In addition, the role of the gut−brain axis is increasingly recognized in the pathogenesis of both the conditions and an alteration of the microbiome is supposed to be associated with an increased risk of mental disorders [19]. On the other hand, the chronic course of IBD, the related stigma and social burden may favor the development of depressive and anxiety symptoms [20].

The gender differences in the prevalence and clinical presentations of both IBD and affective disorders are explained by the different regulation and action of sex hormones and neuropeptides in males and females [21,22,23]. In relation to IBD, females have a higher risk of CD in early adulthood and old age than males, while the opposite occurs for UC [24]. Regarding mental health, women are two to three times more likely to have major depression and anxiety disorders than men [25,26]. In addition, women with major depression are more vulnerable to metabolic abnormalities [25], while IBD symptoms worsen during the menstrual cycle, independently of disease activity [27]. In light of these data, clinicians should consider gender when managing patients with IBD and psychiatric comorbidities [28]. 

The aim of this article is to offer an overview of gender differences in the frequency and presentation of affective disorders in IBD patients in order to help clinicians provide more individualized treatment in the context of precision medicine. 

## 2. Materials and Methods

A search was performed in PubMed (National Library of Medicine) to identify relevant papers. All original articles written in English from 1 January 2013 to 31 December 2022 that had abstracts and full texts were included. Two authors subsequently checked and extracted the following data from the included articles: authors and title, year of publication, characteristics of the study (country and sample size), and information relevant to this overview. The search was performed using the following keywords: “Inflammatory Bowel Disease”, “CD”, “UC”, “mental disorders”, “depression”, and “anxiety”.

Inclusion criteria were (1) original articles, (2) mean age of patients ≥ 18 years, (3) presence of information on anxiety and depression in IBD patients, and (4) the article focused on gender differences in affective disorders in patients with IBD.

Exclusion criteria were (1) reviews, meta-analyses, commentaries, letters, case reports, pooled analyses, comments, case studies, and study protocols; (2) mean age of patients < 18 years; (3) studies conducted on animals; (4) studies on psychiatric conditions other than anxiety or depressive disorders; (5) studies not examining the present topic; and (6) articles not written in English.

## 3. Results

### 3.1. Characteristics of the Included Studies

Most of the included studies had a retrospective design [29,30,31,32,33,34], three were cross-sectional [35,36,37], and three were prospective [38,39,40]. 

The sample sizes and the type of patient (e.g., inpatients or in clinical remission) varied greatly across studies. Tarar et al. [32] investigated the largest sample of IBD patients (nearly 1 million subjects).

The methods used to assess anxiety and depressive disorders differed across the studies; most used the HADS [35,37,38,39,40] or the ICD criteria [29,30,31,32,33,34], only Al-Aamri et al. assessed depression using PHQ-9 [36].

### 3.2. General Considerations

More than 1000 articles were identified by the initial search, of which 12 met all of the criteria and were included. These articles showed that the prevalence of depressive disorders in IBD patients (without stratification by gender or type of IBD) ranged from 13–14% [29] to 50% [35], while the prevalence of anxiety disorders ranged from 11% [30] to 42% [38]. With the exception of two studies [31,39], all included articles reported a significantly higher frequency of depressive or anxiety disorders in females versus males. 

A summary of the results of the main findings of the included studies are reported in Table A1 (Appendix A).

### 3.3. Gender Distribution of Affective Disorder in IBD Patients 

#### 3.3.1. Depressive Disorders 

The following methods were used to assess the presence of depressive disorders: diagnostic criteria from the International Classification of Diseases (ICD) or cut-off scores of rating scales such as the Patient Health Questionnaire (PHQ-9) [41] or the Hospital Anxiety and Depression Scale (HADS) [42]. 

A recent cross-sectional study [36] reported that 22.3% of their total sample had moderate to severe depression, corresponding to a PHQ-9 score ≥ 12. In addition, the frequency of clinically significant depressive symptoms in females was twice that in males. Similar results were reported by a European study by Sciberras et al. in IBD patients in clinical remission [38]. In contrast, a retrospective study conducted in the USA by Ali et al. found lower rates of depressive disorders in hospitalized patients (13.4% for UC and 14.9% for CD), but this was assessed using the ICD criteria for depressive disorders. The authors confirmed the higher risk of depression in women than in men [29]. Another American study by Tarar et al. reported similar rates of depressive disorders in patients hospitalized for IBD, using ICD criteria, and confirmed the higher vulnerability to depression in females than in males [32]. These results confirm the findings of two previous American studies of IBD patients without a previous psychiatric diagnosis [30,33]; the study by Ananthakrishnan et al. examined the risk of depressive disorders following bowel resection surgery. Finally, female gender and perianal surgery were identified as factors correlated with the presence of depressive disorders in an Italian study by Maconi et al. [40].

Four studies from Turkey, Canada, Switzerland, and Taiwan [31,35,37,39] did not find significant differences in the risk of developing depressive disorders between females and males. 

#### 3.3.2. Anxiety Disorders

A recent retrospective study from the USA reported a high prevalence (12.0%) of generalized anxiety disorder (GAD) in IBD inpatients, with females having a higher risk than males [34]. Similar results were found by a recent American study on anxiety disorders [32]. Sciberras et al. [38] reported even higher rates of anxiety disorders in European patients in clinical remission (41.8%), with women having approximately twice the risk as the men. Although they used the same HADS scale as Sciberras et al. for the detection of anxiety disorders, three other studies found lower frequencies of anxiety in IBD patients, but confirmed the higher rates in women compared with men [35,37,40]. In an American study, female patients who had undergone bowel resection surgery had a higher risk of anxiety disorders compared with men [30].

Regarding depressive disorders, the Canadian study by Narula et al. [39] failed to find any gender difference in the risk of developing anxiety disorders in IBD patients.

## 4. Discussion

The findings of the included studies confirm that depressive and anxiety disorders are more prevalent in IBD patients than in the general population [26], so these represent subjects who are vulnerable to suffer from psychiatric conditions. Of note, comparable to the situation in the general population [26], women with IBD seem to have an higher risk of having depressive or anxiety disorders than men. In this regard, a recent meta-analysis by Barberio et al. reported a pooled prevalence of 33.8% for anxiety symptoms in females and 22.8% in males, with an odds ratio (OR) of 1.7 [95% confidence interval (CI) 1.2–2.3]. Similarly, for depressive symptoms, the pooled prevalence was 21.2% in females and 16.2% in males, with an OR of 1.3 [95% CI 1.0–1.8] [21]. These observations were similar in patients with other inflammatory conditions such as atopic dermatitis, rheumatoid arthritis, cardiovascular diseases, and diabetes [43,44,45,46].

This means that, independently of the susceptibility to affective disorders conferred by IBD, gender-related factors increase vulnerability to mood and anxiety disorders in females compared with males [25]. 

The findings of the included studies in this review demonstrate a remarkably high rate of depressive disorders in women with IBD (about 65%) [29,32,33,36]. The study with the largest sample reported the greatest prevalence of depressive disorders in female inpatients (68%) [32]. Similar data were reported for anxiety disorders [32,34]. Taken as a whole, the different authors agree that patients with IBD have a higher probability to develop affective disorders compared with the general population and that female gender represents a risk factor. The differences in the findings of the studies (especially in relation to the frequency of depressive/anxiety disorders) may depend on several factors such as the severity of IBD (e.g., inpatients versus outpatients) [47], the type of IBD (CD versus UC), the method used to diagnose depressive and anxiety disorders, the type of drug treatment (e.g., corticosteroids) [48], and cultural/psychosocial aspects affecting access to mental health support and thus the possibility of an early diagnosis [49]. As also found by Neuendorf et al. [18], the studies included in this overview reported a higher frequency of affective disorders in patients with CD than with UC [29,30]. Regarding the methods used for assessment, the PHQ-9 is a questionnaire characterized by high sensitivity in identifying depressive symptoms and is frequently used in general medical populations (especially in primary care settings) due to its versatility and the quick completion time for patients [50]. On the other hand, HADS was originally designed to assess depressive and anxiety symptoms in hospitalized patients [42]. It is important to underline that both instruments are used to evaluate the severity of psychiatric symptoms and that cut-off scores are used to indicate the probability of a formal psychiatric diagnosis [51]. PHQ-9 was administered only in one study [36] that used a very strict cut-off score (≥12) for the detection of depressive disorders (a score ≥ 10 is generally advised to discriminate the absence or presence of depressive disorders). The study that included subjects with overlapping clinical characteristics (outpatients) and used HADS [38] reported similar frequencies of clinically significant depressive symptoms (22.3% versus 24.5%). 

There are different hypotheses regarding the gender differences in the prevalence of affective disorders in IBD patients [52,53]. An area of growing interest is the gut–brain axis, which may be involved in the pathogenesis of anxiety and depressive disorders in IBD [54]. A recent systematic review highlighted how different levels of sex hormones (testosterone and oestradiol) can affect the diversity and composition of the microbiome, thus explaining the gender differences in the frequency of affective disorders in IBD patients [55]. Of note, recent findings highlight that women and men have a different microbiome composition, thus the eventual complementary administration of probiotics for the treatment of both IBD and affective disorders could be individualized according to the patient’s sex [56]. 

In addition, in the intestinal epithelium, especially in the colon, oestrogen receptor (ER) β plays an important role in the integrity of tight junctions and the barrier function [57]. Of note, some authors reported a significant reduction in ERβ expression in active IBD [57]. A study by Ma et al. [58] demonstrated that ERβ knockout mice developed both colitis and anxiety-like behaviors after the administration of dextran sodium sulfate (an agent that induces acute colitis). Anxiety and depressive disorders may also be exacerbated by impaired regulation of the hypothalamic–pituitary–adrenal (HPA) axis by gonadal hormones including oestrogens [59]. Oestradiol is protective against psychiatric disorders because it strengthens the response of the HPA axis to stressors [60], inhibits cell-mediated immunity associated with depressive disorders [61,62], and exerts antidepressant effects by blocking N-methyl-D-aspartate (NMDA) glutamatergic receptors in the central nervous system [63]. These factors explain the greater predisposition of women to affective disorders compared with men, in particular during specific phases of life such as the post-partum period [64] or menopause [65]. Similarly, women may experience exacerbation of IBD during menstruation [27]. There is a complex interaction between gonadal hormones, affective disorders, gut microbiota, and IBD [66], and it is no coincidence, for example, that CD is associated with depressive disorders, and is more frequent in women than in men in old age [24].

In addition to molecular mechanisms, a number of psychosocial and clinical factors may also explain gender differences in vulnerability to affective disorders among IBD patients. Different authors have reported that women with IBD generally have a worse perception of their quality of life compared with men [67]. This is explained by the fact that nearly half of women experience vulvovaginal discomfort and related sexual dysfunction during IBD flares [68,69], and these symptoms may be particularly severe during menopause [70]. Of note, sexual dysfunction was identified as a predictor of depressive symptoms in IBD patients [71]. In addition, women might experience more distress because many can actively avoid pregnancy for fear of the adverse effects on the fetus related to disease activity and treatments [72]. Finally, compared with male patients, female subjects report more concerns about their body image and function [73,74], which can trigger depressive disorders [75].

Gender stereotypes may also contribute to the gender differences seen in the frequency of affective disorders [76]. Women are more likely than men to utilize mental healthcare services [77,78] and clinicians might be more likely to investigate depressive and anxiety disorders in women than in men [79]. Finally, preliminary data indicate that female healthcare professionals have a more positive attitude towards patients with mental health conditions than male healthcare professionals [80]. 

Finally, it is interesting to highlight that also in pediatric patients affected by IBD, the prevalence of affective disorders is higher compared with the healthy population [81,82] although there are less data than for the adult population. A study by Glapa-Nowak et al. showed that females reported more anxiety symptoms and pain compared with males during disease flares [82]. This aspect, again, could be due to psychosocial aspects or to biological factors such as a different exposition of brain to male hormones during pregnancy according to the sex of the fetus [83].

In conclusion, women with IBD are more likely than men to have an affective disorder and may require specific management involving different healthcare professionals. Future research is needed to confirm these findings as the included studies had the following limitations: (1) they used different tools to assess depressive and anxiety disorders; (2) included patients had different severities of illness or different diagnoses (CD or UC); (3) the study designs differed; (4) there is not a clear distinction between sex and gender; and (5) the studies were conducted in countries with different healthcare systems and cultural attitudes that may influence the diagnosis of affective disorders [84]. 

## 5. Conclusions and Future Directions

The findings reported in the present manuscript support that gender can have implications for the management of IBD. Of note, women with IBD are more likely to develop affective disorders than men. This implies that female patients may need to be treated with different drugs and require specific monitoring and a targeted multidisciplinary approach. Individualized treatment for women is even more important as the presence of affective disorders is associated with a poorer prognosis in IBD [34] and quality of life [38]. Healthcare professionals should regularly assess the mental health of their patients, both during visits and remotely [85], in light of the availability of easily used tools and the detrimental effects of poor psychological wellbeing on the course of illness [86]. A biopsychosocial, multidisciplinary, and individualized approach should thus be promoted to improve prognosis [87]. This is particularly important for inflammatory diseases that share pathogenetic aspects with affective disorders and whose prognosis is strongly influenced by the presence of psychiatric disorders [88]. This is also demonstrated by preliminary data which indicate that the new biologic disease-modifying anti-inflammatory drugs not only influence the course of inflammatory diseases, but improve the psychological wellbeing of patients [89]. 

## Data Availability

The datasets generated during and analyzed during the current study are not publicly available; however, they are available from the corresponding author upon reasonable request.

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
