# Peer review of "Depressive and Anxiety Disorders in Patients with Inflammatory Bowel Diseases: Are There Any Gender Differences?"

_ijerph, 2023, doi:10.3390/ijerph20136255_

Round 1

Reviewer 1 Report

Line 24 „should be limited” I think this cannot be omitted or limited. I suggest “should be thoroughly reconsidered”

Consider whether the study could be reanalyzed as meta-analysis?

I think the study lack novelty as there is a recent systematic analysis with meta-analysis published in Lancet Gastroenterol Hepatol in 2021 (inception to Sept 30, 2020). Although the paper does not strictly focus on the gender differences, there is a significant insight on that (e.g. “Overall, women with IBD were more likely to have symptoms of anxiety than were men with IBD (pooled prevalence 33·8% [95% CI 26·5-41·5] for women vs 22·8% [18·7-27·2] for men; OR 1·7 [95% CI 1·2-2·3]). They were also more likely to have symptoms of depression than men were (pooled prevalence 21·2% [95% CI 15·4-27·6] for women vs 16·2% [12·6-20·3] for men; OR 1·3 [95% CI 1·0-1·8]).”). This study should be discussed in more detail in the discussion section.

However the reviewed study adds two more years (till December 31th 2022). Maybe adding clinical characteristics would enrich the findings.

The discussion seem to be well written and takes into account the bias caused by different rate in usage of mental services (lines 205-210). Consider adding a paragraph on gender differences in anxiety in pediatric population (e.g. Glapa-Nowak A, Subjective Psychophysical Experiences in the Course of Inflammatory Bowel Disease—A Comparative Analysis Based on the Polish Pediatric Crohn’s and Colitis Cohort (POCOCO). International Journal of Environmental Research and Public Health. 2021; 18(2):784.)

.

Reviewer 2 Report

Suggestion: Minor revisions

The authors have done a good job of keeping the paper to the point with sufficient details covering majority of the aspects impacting depression/anxiety in IDBs. Here are some minor suggestions to further strengthen their article.

1.       Please shift section 3.3 to 3.1, 3.1 to 3.2 and 3.2 to 3.3 as current 3.3 provides an overview of the characteristics of all the studies included; such that the sequence is: characteristics  followed by à general considerations followed by à gender distribution. This will help section 3 overall flow better.

2.       Line 147-148, In addition to your example of atopic dermatitis, please add 3 to 4 more inflammatory conditions (like rheumatoid arthritis etc) and the evidence of published impacted mental health as their reference.

3.       Please re-phrase  conclusion as ‘conclusion and future directions’ and add another paragraph on how your work provides evidence towards the much needed attention in mental health, not just in IBS but also, as you rightly mentioned in inflammatory disorders in your discussion.

Reviewer 3 Report

Dear authors,

thank you for the fascinating study. You have done a great job compiling information about inflammatory bowel disease and depression and anxiety disorders. Please find bellow specific comments and suggestions that will improve the quality of this manuscript.

Dear authors,

I apologize for not separating my comments and suggestions and the quality of English language. They are all included in the comments and suggestions section.

Round 2

Reviewer 1 Report

I accept in present form